# Online Left-Hemispheric In-Phase Frontoparietal Theta tACS Modulates Theta-Band EEG Source-Based Large-Scale Functional Network Connectivity in Patients with Schizophrenia: A Randomized, Double-Blind, Sham-Controlled Clinical Trial

**DOI:** 10.3390/biomedicines11020630

**Published:** 2023-02-20

**Authors:** Ta-Chuan Yeh, Cathy Chia-Yu Huang, Yong-An Chung, Sonya Youngju Park, Jooyeon Jamie Im, Yen-Yue Lin, Chin-Chao Ma, Nian-Sheng Tzeng, Hsin-An Chang

**Affiliations:** 1Department of Psychiatry, Tri-Service General Hospital, National Defense Medical Center, Taipei 114202, Taiwan; 2Department of Life Sciences, National Central University, Taoyuan 320317, Taiwan; 3Department of Nuclear Medicine, College of Medicine, The Catholic University of Korea, Seoul 07345, Republic of Korea; 4Department of Psychology, Seoul National University, Seoul 08826, Republic of Korea; 5Department of Emergency Medicine, Tri-Service General Hospital, National Defense Medical Center, Taipei 114202, Taiwan; 6Department of Emergency Medicine, Taoyuan Armed Forces General Hospital, Taoyuan 325208, Taiwan; 7Department of Psychiatry, Tri-Service General Hospital Beitou Branch, National Defense Medical Center, Taipei 112003, Taiwan

**Keywords:** transcranial alternating current stimulation, frontoparietal theta coupling, schizophrenia, negative symptoms, electroencephalography, functional connectivity

## Abstract

EEG studies indicated that schizophrenia patients had increased resting-state theta-band functional connectivity, which was associated with negative symptoms. We recently published the first study showing that theta (6 Hz) transcranial alternating current stimulation (tACS) over left prefrontal and parietal cortices during a working memory task for accentuating frontoparietal theta-band synchronization (in-phase theta-tACS) reduced negative symptoms in schizophrenia patients. Here, we hypothesized that in-phase theta-tACS can modulate theta-band large-scale networks connectivity in schizophrenia patients. In this randomized, double-blind, sham-controlled trial, patients received twice-daily, 2 mA, 20-min sessions of in-phase theta-tACS for 5 consecutive weekdays (n = 18) or a sham stimulation (n = 18). Resting-state electroencephalography data were collected at baseline, end of stimulation, and at one-week follow-up. Exact low resolution electromagnetic tomography (eLORETA) was used to compute intra-cortical activity. Lagged phase synchronization (LPS) was used to measure whole-brain source-based functional connectivity across 84 cortical regions at theta frequency (5–7 Hz). EEG data from 35 patients were analyzed. We found that in-phase theta-tACS significantly reduced the LPS between the posterior cingulate (PC) and the parahippocampal gyrus (PHG) in the right hemisphere only at the end of stimulation relative to sham (*p* = 0.0009, corrected). The reduction in right hemispheric PC-PHG LPS was significantly correlated with negative symptom improvement at the end of the stimulation (r = 0.503, *p* = 0.039). Our findings suggest that in-phase theta-tACS can modulate theta-band large-scale functional connectivity pertaining to negative symptoms. Considering the failure of right hemispheric PC-PHG functional connectivity to predict improvement in negative symptoms at one-week follow-up, future studies should investigate whether it can serve as a surrogate of treatment response to theta-tACS.

## 1. Introduction

Schizophrenia is a chronic, debilitating mental illness. Its core components include positive symptoms, negative symptoms, and cognitive deficits. Primary negative symptoms (alogia, asociality, amotivation, anhedonia, and affective flattening) account for a large part of poor functional outcomes and long-term disability in patients with schizophrenia [1]. Efficacious treatments for negative symptoms remain an important unmet need in schizophrenia.

Recent research indicates that transcranial electrical stimulation (tES) is a representative non-invasive brain stimulation method that may serve as an effective therapy in counteracting negative symptoms [2]. Common tES approaches are transcranial direct current stimulation (tDCS), transcranial alternating current stimulation (tACS), and transcranial random noise stimulation (tRNS). Our earlier research demonstrated the efficacy of adjunct tDCS and tRNS over the prefrontal cortex in improving negative symptoms of schizophrenia [3,4]. Results of a small case series indicated that tACS applied at theta frequency over the prefrontal cortex and delivered during a resting state can improve negative symptoms of schizophrenia [5]. tACS applies a low-intensity (0.5 to 4 mA) sinusoidal (frequency: 0.1 to 80 Hz; phase: 0–360 degrees; DC offset: with/without) electrical currents to the targeted brain regions through the specific electrode montages on the scalp. tACS can enhance local oscillatory activity (i.e., an increase in the power of endogenous brain oscillations in the range of the stimulated frequency) and entrain (i.e., an alignment of local brain endogenous oscillations and the tACS phase) the brain rhythms of the targeted brain region [6]. The sustained aftereffects of a single session of tACS can last up to 70 min after stimulation [7].

Functional magnetic resonance imaging (fMRI) studies identified working memory (WM) related activities in a widely distributed neural network involving the dorsolateral prefrontal cortex (DLPFC), medial prefrontal cortex (mPFC), insula, hippocampus, posterior parietal cortex (PPC), and occipito-temporal cortex [8]. Data from simultaneous fMRI-EEG scans during the retention period of a WM task demonstrated negative correlations between the EEG theta (5–7 Hz) frequency band and the blood oxygen level dependency (BOLD) signal, as well as a strong anatomical overlap in the default mode network (DMN) for the BOLD activation pattern and theta EEG-BOLD signal correlations, which were not present in other frequency domains [9]. A growing body of evidence suggests theta (~6 Hz) phase synchronization (i.e., theta coupling at ~0° relative phase) of distant cortical areas (e.g., frontoparietal regions) as vital neural mechanisms in WM performance, requiring the large-scale integration of functional cortical circuits [10]. For example, research indicated a significant increase of theta (~6 ± 1 Hz) phase synchronization between left DLPFC and PPC occurring ~200–500 ms after memory probe onset while the participants performed a WM task [10]. Polania et al. used tACS simultaneously applied at 6 Hz over the left prefrontal and parietal cortices with a relative 0° (‘‘synchronized’’ or “in-phase” condition) or 180° (‘‘desynchronized’’ or “anti-phase” condition) phase difference versus the placebo stimulation, while healthy individuals performed a working memory task (i.e., online stimulation) [10]. The results indicated that as compared to the placebo stimulation, exogenously induced frontoparietal theta synchronization by in-phase theta-tACS significantly improved cognitive performance, and exogenously induced frontoparietal theta desynchronization by anti-phase theta-tACS deteriorated performance. In our recent double-blind randomized sham control trial, we applied in-phase tACS tuned at theta frequency (6 Hz) and delivered this while the participants performed a working memory task (online theta-tACS), aiming to artificially induce coupling behaviorally relevant brain rhythms between left frontal and parietal regions. The results indicated that 10 sessions of online theta-tACS significantly improved negative symptoms (primary outcome) as well as working memory performance (secondary outcome) in stabilized patients with schizophrenia [11]. It remains unknown whether the repeated application of online theta-tACS can alter or modulate resting-state cortical networks in patients with schizophrenia.

Research indicates that a fundamental pathophysiological feature of schizophrenia consists of a functional disconnection syndrome, in which neural oscillations at the theta frequency are robustly implicated [12]. For example, evidence from studies of patients with schizophrenia indicates increased theta activity during resting conditions and reduced activity for targets during active paradigms [13]. Excessive theta activity (4–8 Hz) during resting conditions is involved in impaired cognitive processes (e.g., working memory, processing speed, verbal learning, reasoning, and problem-solving) [14,15] and negative symptom burden [16] in patients with schizophrenia. Research that performed EEG-based source localization analyses to investigate the resting-state neural connectivity in patients with schizophrenia as compared to healthy controls reported increased functional connectivity between brain networks in the theta frequency [17,18] and an increase in the synchronization between the posterior cingulate cortex, cuneus, and precuneus [19]. Moreover, the increased theta-band functional connectivity between brain networks was associated with a delay in the initiation of information processing [19] and a worse processing speed and verbal memory [17,18]. A recent study of drug naïve first-episode psychosis patients indicated that theta-band connectivity between right frontal and parietal regions underlying the active detection of auditory novelty stimuli during an auditory involuntary attention task was inversely correlated with negative symptoms [20].

Motivated by the above-mentioned findings, suggesting long-range EEG phase synchronization in a narrow theta band of 5–7 Hz as an underlying neural mechanism of WM related activities [10], this study aimed to investigate whether applying 10 sessions of in-phase theta (6 Hz)-tACS for externally inducing left frontoparietal synchronization during a WM task would modulate the resting-state theta-band (5–7 Hz) EEG source-based large-scale functional network connectivity in patients with schizophrenia and determine if the characteristics of network connectivity serve as predictors or surrogates of treatment response. We did not construct specific hypotheses regarding the direction of changes in theta-band network connectivity due to the paucity of relevant previous studies.

## 2. Materials and Methods

### 2.1. Participants

The present study is a double-blind, randomized, sham-controlled trial, approved by the ethics committee of Tri-Service General Hospital, Taipei, Taiwan and registered (ClinicalTrials.gov (accessed on 15 February 2023), ID: NCT04545294; Appendix A). The inclusion criteria were: (1) patients aged 20–65 with DSM-5-defined schizophrenia or schizoaffective disorder; (2) duration of illness >2 years; (3) being clinically stable and on an adequate therapeutic dose of antipsychotics for at least 8 weeks before enrolment; (4) agreement to participate in the study and provide written informed consent. The exclusion criteria were (1) unstable medical conditions, current psychiatric comorbidity, prominent mood symptoms, or active substance use disorders (with the exception of caffeine and/or tobacco); (2) a history of seizures, meningitis, or encephalitis; (3) contraindications for transcranial electrical and magnetic stimulation; (4) a history of intracranial neoplasms or surgery, or a history of severe head injuries or cerebrovascular diseases; (5) pregnancy or breastfeeding at enrollment; (6) scalp skin lesions at the area of electrode application.

### 2.2. Study Design

This randomized, double-blind, and sham-controlled clinical trial was performed between August 2019 and April 2020. Each participant was randomly assigned to one of two groups (i.e., tACS: sham = 1:1). A study coordinator not involved in the trial created 5-digit random numbers and assigned the randomization numbers to the participants to maintain blinding integrity (Appendix A). The participants and researchers were unaware of group assignment until the unblinding of the trial. The primary outcome, the change in the negative symptom severity over time, was measured by the negative symptom subscale score of the Positive and Negative Syndrome Scale (PANSS) at baseline, end of stimulation, and at 1-week and 1-month follow-up visits (Appendix A). Detailed analyses of clinical data have been reported in a separate paper [11].

### 2.3. Brain Stimulation

Online in-phase theta-tACS was applied during the performance of a dual n-back task involving both visuospatial and auditory-verbal WM paradigms (see Appendix A). Sinusoidal tACS was delivered by two battery-operated devices (Eldith DC stimulator Plus, NeuroConn, Ilmenau, Germany) connected with two 4 × 1 wire adaptors (Equalizer Box, NeuroConn), via 10 carbon rubber electrodes (1 cm radius, high-density 4 × 1 configuration with a gel layer of 2.0 mm), at 6 Hz frequency, 2 mA current intensity without DC offset, with 100 cycles ramp-up/ramp-down and a 0° relative phase, for 20 min, twice-daily on 5 consecutive weekdays. In the sham group, short continuous currents without neuromodulatory effects (i.e., 2 mA normal-like stimulation for the initial 30 s followed by a tiny current pulse of 110 μA over 15 ms for impedance control taking place every 550 ms during the remaining time) were applied to mimic real-stimulation sensations. Stimulation electrodes of one DC stimulator were placed at the international 10-10 electrode position F1, F5, AF3, and FC3 while the return electrode was placed at CPz. Stimulation electrodes of the other DC stimulator were placed at P1, P5, CP3, and PO3 while the return electrode was placed at FCz (Appendix A). These electrodes were held in place with conductive paste to keep the combined impedance of all electrodes <15 kΩ. A custom-made pulse generator controlled the two stimulators to achieve a synchronous (in-phase) setup (0° relative phase difference between the output signals of the two stimulators). The chosen montage was adapted from Polania et al. [10] for theta-tACS to externally induce left frontoparietal synchronization. HD-Explore^®^ (Soterix Medical, New York, NY, USA) was used to simulate the numerical computation of the electric field of the current montage (Figure 1).

### 2.4. Electrical Source Estimation of Resting-State EEG

Resting-state EEG (rsEEG) was collected at baseline, end of stimulation, and one-week follow-up, using a 32-channel EEG cap (NP32, GmbH, Ilmenau, Germany) with Ag/AgCl sintered ring electrodes placed according to the international 10–20 system and referenced to the tip of the nose, together with Neuro Prax^®^ TMS/tES compatible full band DC-EEG system (NeuroConn GmbH, Ilmenau, Germany) with a sampling frequency of 4000 Hz, an analogue-digital precision of 24 bits, and an analogous bandpass filter (0–1200 Hz). Patients were seated comfortably in a recliner in a light and sound-attenuated room. They were instructed not to drink caffeinated beverages one hour prior to, and alcohol 24 h prior to EEG recording to avoid caffeine- or alcohol-induced changes in the EEG stream. The ground electrode was placed at Fpz. The horizontal electrooculogram (HEOG) was recorded by two electrodes placed at 1 cm from the outer canthi of both eyes. Two electrodes were placed above and below the left eye, respectively, to record the blinks and vertical electrooculogram (VEOG). The impedance of each electrode was checked to remain below 5 kΩ. Before starting the EEG recording, the patients performed 3-min calibration tasks to estimate the influence of horizontal/vertical movements and blink artifacts on EEG, which was processed and stored in the Neuro Prax^®^ EEG system built-in software providing fully automatic correction of real-time EEG artifacts caused by blinking and eye or body movement during the subsequent EEG recording (Appendix A). rsEEG with eyes open (5 min) and eyes closed (5 min) were recorded for a total of 10 min and the sequence was randomized and counterbalanced across the patients. Patients were instructed to visually fixate on a crosshair in front of them during the eyes-open condition or stay relaxed in a state of mind wandering (i.e., without goal-oriented mental activity) with their eyes closed during the eyes-close condition. EEGLAB v2020.0 [21], an interactive Matlab (MathWorks, Natick, MA, USA) toolbox was used for preprocessing the offline EEG data, e.g., data being downsampled to 500 Hz, band-pass filtered to 1–100 Hz with the finite impulse response method and analog 60 Hz-notch filtered. Bad channels were automatically detected and removed based on artifact subspace reconstruction (ASR) [22]. Independent component analysis (ICA) followed by ICLabel [23] was used to automatically remove artifacts caused by muscle activity, heartbeats, eye movements, and eye blinks (see Appendix A). Considering that theta-tACS was applied in an eyes-open state, only accepted epochs of eyes-open EEG data collected in a resting state were selected for electrical source estimation. All participants had one artifact-free epoch with a length of a minimum of 60 s, respecting the guidelines from previous research [24].

All source imaging was performed with the exact low-resolution brain electromagnetic tomography (eLORETA), a linear inverse solution used to compute the standardized current density or distribution of current density across voxels in the brain by localizing and reconstructing the intracerebral electrical sources underlying the scalp-recorded activity [24] in a realistic head model [25], using the Montreal Neurological Institute (MNI; Montreal, Quebec, Canada) MNI152 template [26], with the three-dimensional eLORETA inverse solution space (i.e., intracerebral volume) restricted to cortical gray matter and hippocampi, as determined by the probabilistic Talairach atlas [27], and partitioned in 6239 voxels (voxel size 5 × 5 × 5 mm).

### 2.5. Whole-Brain Electrical Source-Based Functional Connectivity

In this study, whole-brain EEG source-based functional connectivity analyses were performed by the ‘‘whole-brain Brodmann areas (BAs)’’ approach, using the anatomical definitions of 84 BAs provided by eLORETA software package and based on the Talairach Daemon “http://www.talairach.org/ (accessed on 15 February 2023)” [28]. Standard electrode positions on the MNI152 scalp were used and the intracerebral volume’s partitioning was restricted to 6239 voxels of 5 × 5 × 5 mm spatial resolution. The electric activity at each voxel in the neuroanatomic MNI space as the exact magnitude of the estimated current density is represented by the eLORETA images. We used Brodmann areas as anatomical labels using MNI space with the correction to Talairach space. We chose a ROI-maker 2 method (available in eLORETA) for the construction of the regions of interest (ROIs). Brain activity was computed with eLORETA in 42 regions, positioning the center in Brodmann Areas (BAs: 1–11, 13, 17–25, 27–47) for each hemisphere (see Appendix A for MNI coordinates of each seed), to obtain a topographic view. Pairs of BAs were analyzed using the values of single voxel with the highest F-ratio value at the centroid of each BA. Analyses of whole-brain functional connectivity were based on lagged phase synchronization (LPS) and were performed by using all 42 BAs in each hemisphere as regions of interest (ROIs) to test interregional functional correlations between any pair of BAs LPS. Thus, connectivity analyses were based on (84 × 83/2=) 3486 pairs of sources distributed throughout the cortex, sufficient for obtaining detailed estimates of connectivity within the constraints posed by the limited spatial resolution of EEG. LPS is a method associated with nonlinear functional connectivity and represents the connectivity of two signals after excluding the instantaneous zero-lag component of phase synchronization caused by intrinsic artifacts or non-physiological effects [24]. A value of 1 indicates perfect synchronization and a value of 0 indicates no synchronization. LPS between 84 ROIs was computed for each artifact-free EEG segment in the frequency domain using normalized Fourier transforms. The data in the theta frequency range (5–7 Hz) were selected for statistical analyses. For additional details see Appendix A.

### 2.6. Statistical Analyses

Statistical analyses were performed using either IBM SPSS Statistics 21.0 software (IBM SPSS Inc., Chicago, IL, USA) or the implemented statistical eLORETA nonparametric mapping (SnPM) tool [24]. Repeated-measures analyses of variance were used to assess the effects of theta-tACS on the PANSS negative symptom subscale score over time, with “time” as the within-group factor and “treatment group” as the between-group factor. When significant “time” × “treatment group” interaction was found, the post-hoc Student’s *t*-tests were used to compare the between-group differences at post-baseline visits. The SnPM analysis tool includes a correction for multiple comparisons. The statistical analyses of between-group changes in theta-band (5–7 Hz) electrical source estimation and source functional connectivity from baseline to post-baseline visits were conducted using t-tests that were corrected for multiple comparisons using a non-parametric permutation procedure (5000 randomizations). Spearman rank correlations were used to analyze the relationships between the significant changes in EEG-based source functional connectivity from baseline to post-baseline visits and treatment response to in-phase theta-tACS. Statistical significance for the results was set at *p* < 0.05 (two-tailed) and the false discovery rate (FDR) was used for multiple comparisons corrections.

## 3. Results

Thirty-six patients were randomly allocated to either receive in-phase theta-tACS (n = 18) or a sham stimulation (n = 18) (Appendix A: CONSORT Flowchart) and all of them completed all 10 sessions of the respective arm. The current study analyzed 35 patients with complete EEG data. Table 1 shows the concise demographics and clinical assessment of the participants. There were no significant differences in the sociodemographic and clinical characteristics at baseline between the theta-tACS group and sham group. A significant group-by-time interaction for the PANSS negative subscale score was found (F = 3.05, *p* = 0.0027). Post-hoc analyses showed significant between-group differences at all post-baseline assessments (Figure 2). The negative symptom severity significantly improved at the end of theta-tACS relative to sham and the effect was maintained at the follow-up visits.

### 3.1. Effects of Theta-tACS on Whole-Brain EEG Source-Based Theta-Band Functional Connectivity

SnPM showed no significant difference in eLORETA seed-based whole-brain analyses of functional connectivity in the theta-band frequency between theta-tACS and sham group at baseline. As can be seen in Figure 3, rows and columns in these matrices represent the 84 ROIs. The element of the matrix was the t value of the independent t test that determines whether the mean change for LPS of any pair of 84 ROIs is significantly different from baseline to postbaseline visits (i.e., end of stimulation and 1-week follow-up) among participants treated with theta-tACS relative to the sham stimulation (i.e., left panels of Figure 3A,B). To avoid ambiguity, the elements of the matrix (i.e., *t* values) greater than the t-value threshold for statistical significance returned 1 and those less than the threshold returned 0 in the map (i.e., right panels of Figure 3A,B). There was a significant between-group change from baseline to end of stimulation in theta-band functional connectivity (only between one pair of ROI 66 and ROI 71, see Figure 3A, right panel). ROI 66 and ROI 71 are posterior cingulate and parahippocampal gyrus, respectively (Appendix A). The between-group change from baseline to one-week follow-up in theta-band functional connectivity was non-significant (Figure 3B, right panel).

To be more specific, the LPS between a region in the posterior cingulate (PC, xyz = 5, −50, 5; BA 29; right hemisphere; Figure 4A) and a region in the parahippocampal gyrus (PHG, xyz = 15, 0, −20; BA 34; right hemisphere) was significantly reduced in theta-tACS group compared with the sham (*p* value = 0.0009, corrected; Figure 4B). The inclusion of the antipsychotic medication dose (in olanzapine equivalents) did not substantially alter the results. In addition, the effects of theta-tACS on functional connectivity were restricted only to the specific theta-frequency (5–7 Hz) band because the results were non-significant for the broader theta band 4.5–7.5 Hz.

### 3.2. Correlation Analyses

The reduction in the LPS between the PC and the PHG from baseline to the end of stimulation was significantly correlated with the improvement of negative symptoms at the end of stimulation (r = 0.503, *p* = 0.039, Figure 5) in theta-tACS group. The inclusion of the antipsychotic medication dose (in olanzapine equivalents) did not substantially alter the results.

## 4. Discussion

To our knowledge, this is the first randomized, double-blind, sham-controlled clinical trial providing evidence that adjunct online in-phase theta (6 Hz)-tACS improves negative symptoms of schizophrenia through modulating theta-band (5–7 Hz) EEG source-based large-scale functional network connectivity. Recent research showed that a single session of online in-phase theta (6 Hz)-tACS with a low current intensity (1 mA) did not result in any significant change in resting-state EEG scalp-level theta-band connectivity between prefrontal and parietal areas in healthy individuals [29]. There was a lack of evidence for the acute and longer-lasting effects of 10 sessions of online in-phase theta (6 Hz)-tACS using a protocol comprising of 2 mA for 20 min (i.e., a total stimulation duration of 200 min) on resting-state EEG source-level theta-band long-range functional connectivity in patients with schizophrenia. In the present study, online in-phase theta (6 Hz)-tACS was applied to externally induce left frontoparietal synchronization during a working memory task. The results of clinical data showed that online in-phase theta (6 Hz)-tACS effectively improved negative symptoms in stable patients with schizophrenia [11] while the EEG results showed that resting-state theta-band (5–7 Hz) functional connectivity between the posterior cingulate and the parahippocampal gyrus in the right hemisphere (Figure 4) was significantly reduced at the end of stimulation among participants treated with in-phase theta (6 Hz)-tACS relative to the sham condition.

Evidence indicates that patients with schizophrenia showed increased EEG-based resting-state functional connectivity at lower frequencies (i.e., hyper-synchronizations in the theta and delta frequencies) compared to the healthy controls, especially higher right fronto-parietal connectivity and higher connectivity between central and parietal areas in the theta band [17], the increased theta-band resting-state connectivity across the midline, sensorimotor, orbitofrontal regions, and the left temporoparietal junction [18], and abnormally strengthened coupling between sources located within the posterior hub of the DMN, containing PC, precuneus, and cuneus [19]. However, these cross-sectional case-control studies cannot determine whether increased theta-band long-range functional connectivity is causal to clinical manifestations of the illness, functionally compensatory processes, or purely epiphenomena.

Our study demonstrated that adjunct 10 sessions of in-phase theta (6 Hz)-tACS during a working memory task resulted in theta-band (5–7 Hz) interregional de-synchronization between two nodes in the posterior part of DMN (i.e., the PC and the PHG) at the end of stimulation along with an improvement in negative symptoms. Furthermore, the reduction in resting-state theta-band connectivity within the above network over time was positively correlated with negative symptom improvement at the end of stimulation, thus supporting the value of right hemispheric PC-PHG LPS as a surrogate biomarker for the clinical efficacy of online in-phase theta-tACS in treating negative symptoms of schizophrenia.

A large number of fMRI/DTI studies indicate that functional hyperconnectivity within the DMN may be the most common finding in comparisons of schizophrenia patients with healthy individuals [30,31] and suggest that increased functional connectivity of the intrinsic networks may be an endophenotype or part of the core pathophysiology of schizophrenia and underlie risks for this serious mental illness [30]. A growing body of evidence also suggests that functional hyperconnectivity within the DMN may play a vital role in the manifestation of negative symptoms of schizophrenia. For example, a previous study indicated that DMN areas with higher activity or connectivity (e.g., frontal polar cortex DMN hyperactivity) contributed to regional functional pathology in schizophrenia and bore significance for negative symptoms [32]. A recent study of neuroimaging coupled with machine learning in patients with schizophrenia-spectrum disorder indicated that resting-state connectivity within the DMN measured at baseline accurately predicted the change in negative symptom severity (accuracy: 83%) at 1-year follow-up, providing its utility in guiding long-term prognostication in schizophrenia [33]. Specifically, the increased connectivity of a left-lateralized sub-network of DMN with direct involvement of hub regions, such as between the posterior cingulate and anterior medial prefrontal cortex (amPFC) and between ventromedial prefrontal cortex (vmPFC) and parahippocampal cortex, predicted the worsening of negative symptoms at 1-year follow-up. A proposed explanation for these findings was based on functional segregation within DMN regions and the putative introspective and extrospective modes of the DMN (i.e., the DMN toggling between an introspective self-referential mode and an extrospective mode that remains alert to changes in the external environment, while normally the internalization and self-reflective thinking under normal conditions are frequently interrupted and shifted toward an externalized state vigilant to external stimuli).

The DMN can be separated into two subsystems (i.e., dorsal medial and medial temporal subsystems), tightly linked by functional hubs also known as a midline core (i.e., amPFC, PC, and angular gyrus, which activate preferentially for self-relevant conditions) [34]. The dorsal medial subsystem comprises the dorsal medial prefrontal cortex (dmPFC), lateral temporal cortex, temporoparietal junction (TPJ) and temporal pole, which activate when subjects reflect on their present mental states. The medial temporal subsystem comprises the vmPFC, hippocampal formation, PHC, retrosplenial cortex, and posterior inferior parietal lobule, which are related to the subjects’ autobiographical memory and future simulations. Hyper-connectivity within the DMN functional hubs or subsystems may contribute to more internalization and fewer shifts to the externalized mode, which just characterizes the internalizing negative symptoms of schizophrenia [33].

Our results suggest a potential relationship between theta-tACS inducing a decrease in right hemispheric PC-PHG functional connectivity in the theta frequency (5–7 Hz) and improvements in negative symptoms, suggesting that theta-band long-range functional hyper-connectivity may be a neural signature of negative symptoms of schizophrenia, and theta-tACS controlled negative symptoms by effectively regulating the PC-PHG connectivity within DMN. It should be noted that a recent fMRI study indicated that hyperconnectivity within the DMN nodes, i.e., the functional connectivity between angular gyrus (AG) and middle temporal gyrus (MTG), was further enhanced in medication-resistant schizophrenia patients receiving a regular course of bitemporal electroconvulsive therapy (ECT) combined with antipsychotics for 4 weeks, as compared to those receiving only antipsychotics, suggesting that ECT did not reverse the original state of hyperconnectivity within the DMN, but provoked a compensatory increase in DMN connectivity [35]. In addition, the increases in functional connectivity between left AG and right MTG and between right AG and right MTG were positively correlated with the reduction in negative symptoms. This study implied that the hyperconnectivity of the DMN in medication-resistant schizophrenia patients reflected a compensatory mechanism to counteract the psychopathological symptoms and also highlighted the need for a separate consideration of stages of illness and various phenotypes. Specifically, evidence indicates that increased functional connectivity or an early compensatory activity is more prominent in early-stage schizophrenia, whereas decreased functional connectivity or decompensation is typical of late-stage schizophrenia [36]. Since our study recruited patients with stabilized schizophrenia and used EEG as a neuroimaging modality that allows for studying neuronal activity in different frequencies. The interpretation of our EEG-based results regarding the theta frequency-specific change of functional connectivity between nodes within DMN after theta-tACS warrants further discussion.

A previous study used magnetoencephalography (MEG) to investigate resting-state brain oscillation and the DMN based on a source space in different frequency bands (i.e., theta, alpha, beta, and gamma) and found that the spatial distribution of DMN activity in the alpha frequency was similar to that reported in fMRI studies, and schizophrenia patients had an increased resting activity of the posterior hub of DMN (i.e., PC), predominantly in the theta band [37]. It is known that theta-band oscillations are generated by the septum pellucidum, medial septal nucleus, and especially the nucleus of the vertical limb of the diagonal band of Broca, which has reciprocal connections with the hippocampal formation (e.g., PHG and surrounding structures such as PC and entorhinal cortex) and is responsible for the generation of theta waves in the brain regions [38]. The PC activity is involved in the processing of information regarding the self (e.g., during any task related to the self and others, remembering the past, and thinking about the future). The activity of the hippocampal formation including PHG is involved in spatial and scene recognition and future simulations. Both PHG and PC play an important role in internalizing negative symptoms of schizophrenia [33] and are also key theta rhythm generators in the brain. The over-synchronization between the two DMN structures may reflect the aberrant and dysfunctional reciprocal connections and represent a primary abnormality underlying the negative symptoms. Taken together, theta-tACS intervention in schizophrenia could be working by desynchronizing the neuronal networks whose over-synchronization accounts for the manifestation of negative symptoms. Future studies are needed to investigate the feasibility of using novel machine learning classifiers (e.g., deep convolutional neural network or fuzzy classifier) for pre- and post-treatment resting-state EEG connectivity to identify schizophrenia patients responding to theta-tACS [39,40].

Our study has a few limitations. First, EEG directly measures fast neural network dynamics with high temporal resolution (ms) for investigating the high temporal dynamics of the whole-brain network functional connectivity but it is limited by its ability to detect sources of electrical activity in deep structures (e.g., thalamus and cerebellum) that were reported to have aberrant connectivity in schizophrenia patients [41], as well as by its spatial resolution recorded using the 32-channel array of scalp electrodes for EEG source localization [42]. Our findings await further investigation using simultaneous EEG-fMRI for the analysis of the brain networks’ functional connectivity [43]. Second, the efficacy of theta-tACS in reducing negative symptom severity was maintained at 1-week follow-up visit, but the between-group difference in the change of right hemispheric PC-PHG functional connectivity disappeared at that timepoint. Therefore, we cannot exclude the possibility that the effect of theta-tACS on right hemispheric PC-PHG functional connectivity was a random finding. We also consider it necessary to exclude the possibility that the effect was purely an epiphenomenon from the impact of theta-tACS on the connectivity of other important large-scale networks implicated in schizophrenia due to the inherent limitations of the eLORETA seed-based approach for functional connectivity analysis with pre-defined seeds or ROIs. Third, it is worth noting that atypical antipsychotics may have a positive impact on the modulation of DMN connectivity [44,45] and thus we cannot exclude the possibility that the changes observed in PC-PHG functional connectivity were mediated by the effects of the interaction between theta-tACS and the antipsychotic medications that the participants were currently receiving. However, the direct effects of antipsychotics seem unlikely in our study since olanzapine equivalent dose had no significant effect on the results. Finally, our study did not include healthy individuals as a control group and thus the interpretation of the statement of theta-band hyper-connectivity in our patients with schizophrenia should be treated with caution. Further studies including both patients and healthy subjects are required to confirm our results.

## 5. Conclusions

Our study indicates that adjunct theta-tACS might contribute to the improvement of negative symptoms by modulating theta-band functional connectivity between spatially separated brain structures within the DMN in patients with schizophrenia. Furthermore, the change in right hemispheric PC-PHG functional connectivity in response to theta-tACS may serve as a biomarker to track the therapeutic effect of this novel non-invasive brain stimulation. Our findings may be a new starting point for the development of individualized and effective brain stimulation therapies in treating negative symptoms of schizophrenia.

## Figures and Tables

**Figure 1 biomedicines-11-00630-f001:**
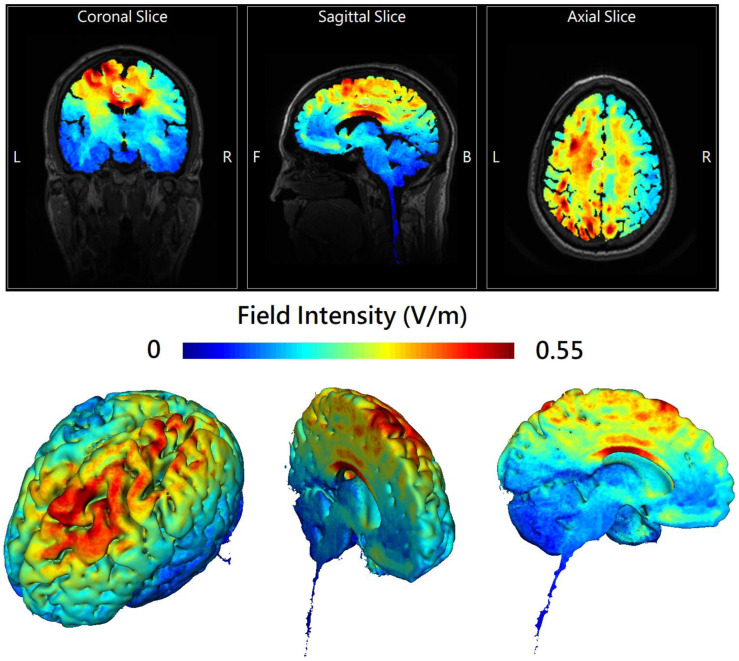
The 2D (**upper** panel) and 3D (**lower** panel) representation of electric field simulation of left-hemispheric frontoparietal theta in-phase transcranial alternating current stimulation (theta-tACS) by HD-Explore^®^ (Soterix Medical, New York, NY, USA), which utilizes a finite element model of brain current flow based on an MRI derived MNI-152 standard brain template. The colorbar indicates the intensity of electrical field (V/m). In the upper panel, the center of the white circle represents the stereotaxic coordinate (x − 11, y − 18, z + 40) where the field intensity is 0.28 V/m.

**Figure 2 biomedicines-11-00630-f002:**
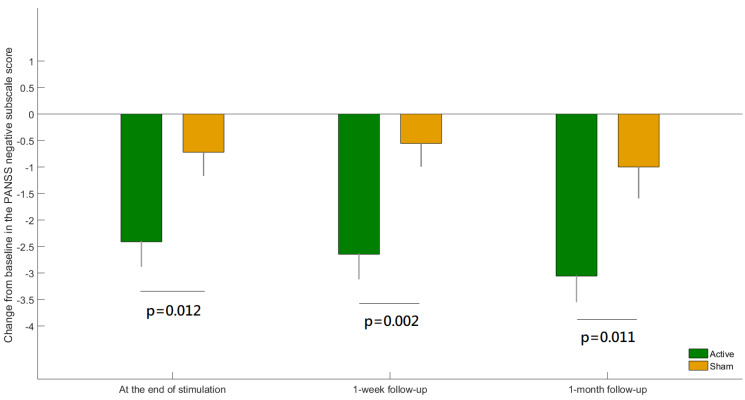
The difference between active versus sham group in the change from baseline for the Positive and Negative Syndrome Scale (PANSS) negative subscale score at each post-baseline visit. Error bars indicate standard errors of the means.

**Figure 3 biomedicines-11-00630-f003:**
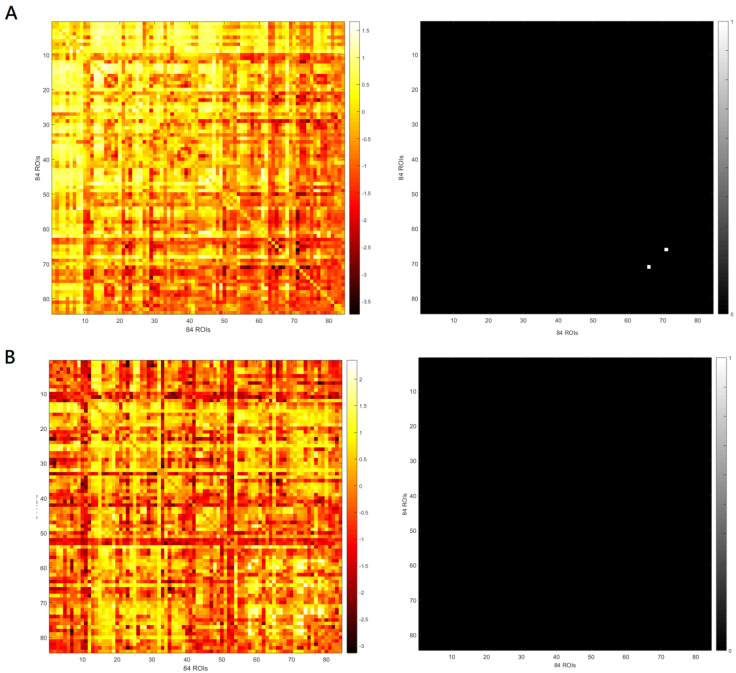
The left panels indicate the matrix of the t value for the mean change for lagged phase synchronization (LPS) of any pair of 84 regions of interest (ROIs) from baseline to the end of stimulation (**A**) and from baseline to the one-week follow-up (**B**). The right panels indicate the maps where *t* values in the matrix of the left panels that are greater than *t*-value threshold for statistical significance return 1 and those less than the threshold return 0.

**Figure 4 biomedicines-11-00630-f004:**
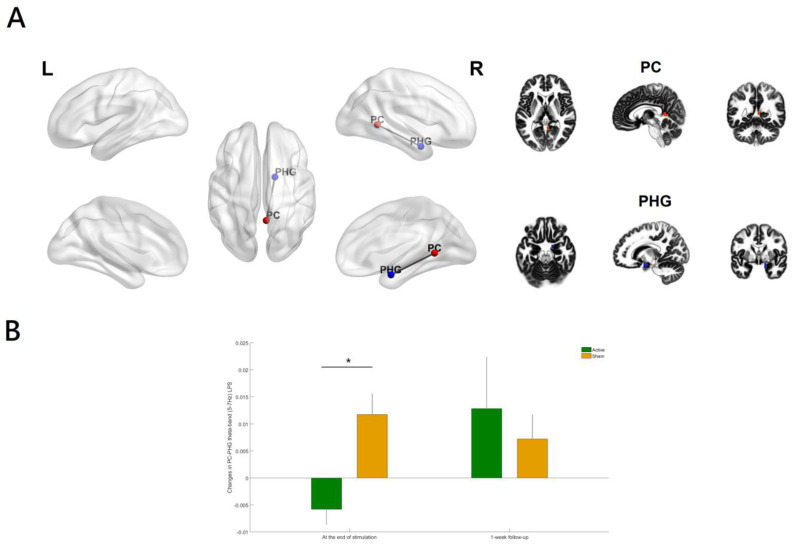
(**A**) Relative to the sham group, the theta-tACS group had a significant decrease in theta-band (5–7 Hz) lagged phase synchronization (LPS) between the posterior cingulate (PC) and the parahippocampal gyrus (PHG) only (**B**) from baseline to the end of stimulation (*p* value = 0.0009, corrected). Color coding: PC (right hemisphere), red; PHG (right hemisphere), blue. The figure was created using eLORETA and BrainNet Viewer. Regions of interest (ROIs) shown here are displayed on a 5 × 5 × 5 MNI template brain in eLORETA for analyses (5 mm resolution is used). Error bars indicated standard errors. * *p* < 0.05 (corrected).

**Figure 5 biomedicines-11-00630-f005:**
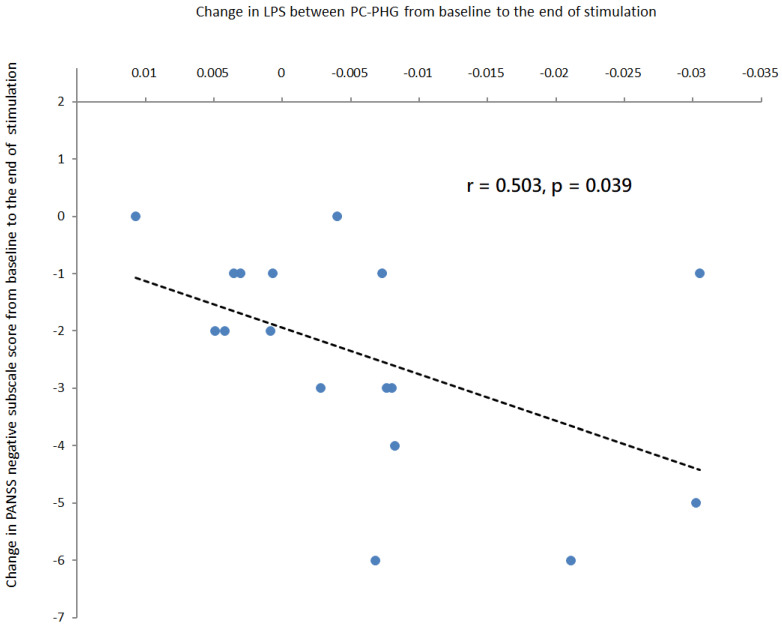
Correlation analyses in patients treated with active online theta-tACS showed that a greater reduction in the LPS between the posterior cingulate and the parahippocampal gyrus from baseline to the end of stimulation was associated with a greater reduction in PANSS negative subscale score from baseline to the end of stimulation. The regression line for the linear regression slope was shown.

**Table 1 biomedicines-11-00630-t001:** Concise demographics and clinical data of the participants.

	tACS (n = 17)	Sham (n = 18)	*p* Value
Schizophrenia/schizoaffective disorder	12/5	15/3	0.44
Gender (f/m)	9/8	8/10	0.62
Handedness (r/l)	16/1	15/3	0.60
Age, years old	42.12 ± 8.99	43.17 ± 11.20	0.76
Years of education	14.6 ± 3.2	12.7 ± 2.8	0.07
Years since diagnosis	15.7 ± 10.6	17.3 ± 10.6	0.65
Olanzapine equivalent dose, mg/day ^a^	19.59 ± 11.83	19.03 ± 13.46	0.90
PANSS total score	71.82 ± 9.64	74.11 ± 7.30	0.43
PANSS negative subscale score	19.00 ± 3.86	19.83 ± 3.63	0.52
PANSS positive subscale score	15.71 ± 5.06	16.28 ± 4.08	0.72
PANSS general subscale score	37.12 ± 5.27	38.99 ± 3.91	0.58
SANS score	50.76 ± 11.10	52.61 ± 10.05	0.61

Abbreviations: tACS, online theta transcranial alternating current stimulation; PANSS, Positive and Negative Syndrome Scale; SANS, Scale for the Assessment of Negative Symptoms. Notes: data are presented as means ± standard deviations unless otherwise stated. ^a^ The daily dose of antipsychotic medications was converted to olanzapine equivalent.

## Data Availability

The data presented in this study are available on request from the corresponding author.

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
