# Peer review of "Online Left-Hemispheric In-Phase Frontoparietal Theta tACS Modulates Theta-Band EEG Source-Based Large-Scale Functional Network Connectivity in Patients with Schizophrenia: A Randomized, Double-Blind, Sham-Controlled Clinical Trial"

_biomedicines, 2023, doi:10.3390/biomedicines11020630_

Round 1
Reviewer 1 Report
This work is interesting. She presents research on the diagnosis of schizophrenia based on the analysis of EEG signals. According to the authors, patients with schizophrenia had an increased functional theta range at rest, which was associated with negative symptoms.
For the study of EEG signals, samples were collected from 35 patients, which were analyzed using special information methods.
In general, the work is presented clearly. However, I would like to propose to slightly extend the analysis of the problem state and pay a little attention to the methods that are used to analyze EEG signals. In particular, I would suggest considering methods based on knowledge discovery, such as classifiers and fuzzy classifiers. For example, such methods were considered in the papers:
Rabcan, J., Levashenko, V., Zaitseva, E., Kvassay, M., Review of methods for EEG signal classification and development of new fuzzy classification-based approach, IEEE Access, 2020, 8, pp. 189720–189734
Lai, C.Q., Ibrahim, H., Suandi, S.A., Abdullah, M.Z., Convolutional Neural Network for Closed-Set Identification from Resting State Electroencephalography, Mathematics, 2022, 10(19), 3442
Author Response
Response to reviewer 1
This work is interesting. She presents research on the diagnosis of schizophrenia based on the analysis of EEG signals. According to the authors, patients with schizophrenia had an increased functional theta range at rest, which was associated with negative symptoms.
For the study of EEG signals, samples were collected from 35 patients, which were analyzed using special information methods.
In general, the work is presented clearly. However, I would like to propose to slightly extend the analysis of the problem state and pay a little attention to the methods that are used to analyze EEG signals. In particular, I would suggest considering methods based on knowledge discovery, such as classifiers and fuzzy classifiers. For example, such methods were considered in the papers:
Rabcan, J., Levashenko, V., Zaitseva, E., Kvassay, M., Review of methods for EEG signal classification and development of new fuzzy classification-based approach, IEEE Access, 2020, 8, pp. 189720–189734
Lai, C.Q., Ibrahim, H., Suandi, S.A., Abdullah, M.Z., Convolutional Neural Network for Closed-Set Identification from Resting State Electroencephalography, Mathematics, 2022, 10(19), 3442
Response:
We would like to thank the reviewer for the insightful comment.
Line 487-491, we add “Future studies are needed to investigate the feasibility of using novel machine learning classifiers (e.g., deep convolutional neural network or fuzzy classifier) for pre- and post-treatment resting-state EEG connectivity to identify schizophrenia patients re-sponding to theta-tACS [39, 40].”
Reviewer 2 Report
The manuscript entitled Online left-hemispheric in-phase frontoparietal theta tACS 2 modulates theta-band EEG source-based large-scale functional 3 network connectivity in patients with schizophrenia: a randomized, double-blind, sham-controlled clinical trial, by Yeh et al, describes the effect of in-phase tACS treatment onto the negative symptomatology schizophrenic patients, comparing with a sham group. The authors claim that in-phase theta-tACS can modulate theta-band large-scale functional connectivity pertaining to negative symptoms while right hemispheric PC-PHG functional connectivity may serve as a surrogate of treatment response. This is an interesting result. The methodology is (in my opinion) as complex as the majority of papers devoted to source location and brain connectivity, but that is the fashion. There are some comments that I would like to clarify before to have the manuscript published.
Major comments
1.- Please, clarify the method to eliminate artefacts, because in line 201-202 it is stated that artefacts were fully automatic corrected by a software built-in the Neuro Prax EEG system, and later they remove the artefacts through ICLabel (line 213).
2.- Why use a so narrow theta band (5-7 Hz), instead a broader band 4.5 – 7.5 Hz.?
3.- eLORETA is based in standardized LORETA (sLORETA) that computes the standardized current density (unitless) and not properly the cortical current density. Please, clarify this point (line 221).
4.- Please, clarify how do you estimates the generators within the theta band, e.g., do you filter the raw EEG record for the band and then use this result to find brain sources?.
5.- I’m not sure if you compute the sources during eyes open or eyes closed states. Please explain. Any way, if you are measuring a total time of 5 mins (eyes open/closed), at what time exactly do compute the brain sources?, do you windowing the record or use the whole trace?.
6.- Although the authors give references for LPS, a scientific article must explain the methods sufficiently, so I think that a succinct explanation of this statistical method (fundamental in the results) should be offered.
7.- What is the meaning of the second decimal for years?, are the authors measuring periods of year-unit with days precision?. In other case, please remove and give only one decimal.
8.- In line 317 and 334 the medication is measured in chlorpromazine equivalents and in table 2 it is indicated olanzapine. Please, explain or remove one of them.
9.- Indicate the statistical test at figure 2.
10.- I cannot understand the timing between electrical stimulation and acquisition of EEG for brain sources. Consider to include a figure with timeline of stimulation, EEG recording and scales during the whole trial.
10.- It is amazing that only one statistical association were observed between tACS and sham patients. Although the significance is quite low, I wonder whether this cannot be a random finding. If this effect is only for end of stimulation, can be an acute effect that vanish before one week? Why did you not provide the results of the one-month follow-up?.
11.- I do not understand what inclusion of antipsychotic medication means (line 316), considering that medication was taken previously by the patients and with similar dosage. I cannot neither comprehend what results were not significant for other bands (lines 319-321), if authors did not specify to have tried other EEG bands at methods.
12.- I’m not convinced that the improving of negative signs would modulated through the EEG theta band. In fact, the negative signs persist lower than sham at one-month follow-up, but the difference in LPS (maybe random, considering the scarce number of regions modified) disappears at one-week follow-up (figure 3). This fact indicates that the decrease of negative symptoms persist even when the LPS in tACS is similar to sham at one-week.
Minor comments
1.- Please, add a list with acronyms.
2.- I cannot find a great relevance to the table 1 and can be moved to supplementary material.
3.- Acronym FDR (line 269) is not defined.
4.- Please, add labels to the figure 4.
Author Response
Response to reviewer 2
The manuscript entitled Online left-hemispheric in-phase frontoparietal theta tACS 2 modulates theta-band EEG source-based large-scale functional 3 network connectivity in patients with schizophrenia: a randomized, double-blind, sham-controlled clinical trial, by Yeh et al, describes the effect of in-phase tACS treatment onto the negative symptomatology schizophrenic patients, comparing with a sham group. The authors claim that in-phase theta-tACS can modulate theta-band large-scale functional connectivity pertaining to negative symptoms while right hemispheric PC-PHG functional connectivity may serve as a surrogate of treatment response. This is an interesting result. The methodology is (in my opinion) as complex as the majority of papers devoted to source location and brain connectivity, but that is the fashion. There are some comments that I would like to clarify before to have the manuscript published.
Major comments
1.- Please, clarify the method to eliminate artefacts, because in line 201-202 it is stated that artefacts were fully automatic corrected by a software built-in the Neuro Prax EEG system, and later they remove the artefacts through ICLabel (line 213).
Response 1
We would like to thank for the reviewer for the insightful comment. We add the methods to eliminate artefacts in the supplementary materials (page 3 for online correction of eye movement and artifacts; page 3-5 for IClabel).
2.- Why use a so narrow theta band (5-7 Hz), instead a broader band 4.5 – 7.5 Hz.?
Response 2
We would like to thank for the reviewer for the important comment. In line 120, we specify a reason for the narrow theta band (5-7 Hz) “Motivated by the above-mentioned findings suggesting that long-range EEG phase synchronization in a narrow theta band of 5-7 Hz as an underlying neural mechanism of WM related activities [10], ”
3.- eLORETA is based in standardized LORETA (sLORETA) that computes the standardized current density (unitless) and not properly the cortical current density. Please, clarify this point (line 221).
Response 3
We would like to thank for the reviewer for the critical comment.
In line 229, we modify the text “All source imaging was performed with the exact low-resolution brain electromag-netic tomography (eLORETA), a linear inverse solution used to compute the standardized current density or distribution of current density across voxels in the brain by localizing and reconstructing the intracerebral electrical sources underlying the scalp-recorded activity… ”
4.- Please, clarify how do you estimates the generators within the theta band, e.g., do you filter the raw EEG record for the band and then use this result to find brain sources?.
Response 4
We would like to thank for the reviewer for the valuable comment.
We did not filter the raw EEG record for the theta band before we applied eLORETA. In line 225-228 and 270-273, we specify this.
5.- I’m not sure if you compute the sources during eyes open or eyes closed states. Please explain. Any way, if you are measuring a total time of 5 mins (eyes open/closed), at what time exactly do compute the brain sources?, do you windowing the record or use the whole trace?.
Response 5
We would like to thank for the reviewer for the valuable comment.
We compute the sources during the eyes open state and use the whole trace of EEG recording. We explain them in line 223-228 “Considering theta-tACS was applied in an eyes-open state, only accepted epochs of eyes-open EEG data collected in a resting state were selected for electrical source estimationpower spectral analysis using fast Fourier transforms to obtain spectral estimates of theta (5–7 Hz) oscillations. All participants had one artifact-free epoch with a length of a minimum of 60 seconds, respecting the guide lines from previous research [24].. ”
6.- Although the authors give references for LPS, a scientific article must explain the methods sufficiently, so I think that a succinct explanation of this statistical method (fundamental in the results) should be offered.
Response 6
We would like to thank for the reviewer for the valuable comment.
We add the relevant information in the Supplement Materials (page 5-6) and in line 266-272 “LPS is a method associated with non-266 linear functional connectivity and represents the connectivity of two signals after excluding the instantaneous zero-lag component of phase synchronization caused by intrinsic artifacts or non-physiological effects [24]. A value of 1 indicates perfect synchronization and a value of 0 indicates no synchronization. LPS between 84 ROIs was computed for each artifact-free EEG segment in the frequency domain using normalized Fourier transforms. The data in the theta frequency range (5-7Hz) were selected for statistical analyses. For additional details see Supplement Materials.”
7.- What is the meaning of the second decimal for years?, are the authors measuring periods of year-unit with days precision?. In other case, please remove and give only one decimal.
Response 7
We would like to thank for the reviewer for the valuable comment.
We remove the year-unit and give only one decimal in Table 1.
8.- In line 317 and 334 the medication is measured in chlorpromazine equivalents and in table 2 it is indicated olanzapine. Please, explain or remove one of them.
Response 8
We would like to thank for the reviewer for the valuable comment.
We correct the error in line 340, 358 and 514. It should be olanzapine throughout the manuscript.
9.- Indicate the statistical test at figure 2.
Response 9
We would like to thank for the reviewer for the valuable comment.
In line 279-285, we add the statistical test for figure 2 “Repeated-measures analyses of variance were used to assess the effects of theta-tACS on PANSS negative symptom subscale score over time, with “time” as the within-group factor and “treatment group” as the between-group factor. When significant “time”× “treatment group” interaction was found, the post-hoc Student’s t-tests were used to compare the between-group differences at post-baseline visits.”
10.- I cannot understand the timing between electrical stimulation and acquisition of EEG for brain sources. Consider to include a figure with timeline of stimulation, EEG recording and scales during the whole trial.
Response 10
We would like to thank for the reviewer for the valuable comment.
We add “ Figure S1. Timeline for treatment and assessments.” in the supplementary materials.
11.- It is amazing that only one statistical association were observed between tACS and sham patients. Although the significance is quite low, I wonder whether this cannot be a random finding. If this effect is only for end of stimulation, can be an acute effect that vanish before one week? Why did you not provide the results of the one-month follow-up?.
Response 11
We would like to thank for the reviewer for the valuable comment.
We did have the EEG data at one-month follow-up (see Figure S1). We provide some explanations for the finding in line 504-510 “Second, the efficacy of theta-tACS in reducing negative symptom severity was maintained at 1-week follow-up visit, but the between-group difference in the change of right hemi-spheric PC-PHG functional connectivity disappeared at that timepoint. Therefore, we cannot exclude the possibility that the effect of theta-tACS on right hemispheric PC-PHG functional connectivity was a random finding. We also consider it necessary to exclude the possibility that the effect was purely an epiphenomenon from the impact of theta-tACS on the connectivity of other important large-scale networks implicated in schizophrenia due to the inherent limitations of the eLORETA seed-based approach for functional connectivity analysis with pre-defined seeds or ROIs”
12.- I do not understand what inclusion of antipsychotic medication means (line 316), considering that medication was taken previously by the patients and with similar dosage. I cannot neither comprehend what results were not significant for other bands (lines 319-321), if authors did not specify to have tried other EEG bands at methods.
Response 12
We would like to thank for the reviewer for the valuable comment.
The reason for the inclusion of antipsychotic medication can be seen in line 513-518 “Third, it is worth noting that atypical antipsychotics may have positive impact on the modulation of DMN connectivity [44, 45] and thus we cannot exclude the possibility that the changes observed in PC-PHG functional connectivity were mediated by the effects of the interaction between theta-tACS and the antipsychotic medications that the participants were currently receiving.” In addition, we delete the results for other bands as suggested.
13.- I’m not convinced that the improving of negative signs would modulated through the EEG theta band. In fact, the negative signs persist lower than sham at one-month follow-up, but the difference in LPS (maybe random, considering the scarce number of regions modified) disappears at one-week follow-up (figure 3). This fact indicates that the decrease of negative symptoms persist even when the LPS in tACS is similar to sham at one-week.
Response 13
We would like to thank for the reviewer for the valuable comment.
We provide some explanations for the finding in line 504-510 “Second, the efficacy of theta-tACS in reducing negative symptom severity was maintained at 1-week follow-up visit, but the between-group difference in the change of right hemi-spheric PC-PHG functional connectivity disappeared at that timepoint. Therefore, we cannot exclude the possibility that the effect of theta-tACS on right hemispheric PC-PHG functional connectivity was a random finding. We also consider it necessary to exclude the possibility that the effect was purely an epiphenomenon from the impact of theta-tACS on the connectivity of other important large-scale networks implicated in schizophrenia due to the inherent limitations of the eLORETA seed-based approach for functional connectivity analysis with pre-defined seeds or ROIs”
Minor comments
1.- Please, add a list with acronyms.
Response 1
Line 561 shows the list with acronyms.
2.- I cannot find a great relevance to the table 1 and can be moved to supplementary material.
Response 2
Table 1 is moved to supplementary materials as Table S2.
3.- Acronym FDR (line 269) is not defined.
Response 3
As suggested, FDR is defined in line 293.
4.- Please, add labels to the figure 4.
Response 4
As suggested, labels are added to figure 4.
Round 2
Reviewer 2 Report
I acknowledge the effort made by the authors in clarify and modify the questions raised.
I think that authors must modify the abstract according to the new paragraph introduce at Discussion: “Second, the efficacy of theta-tACS in reducing negative symptom severity was maintained at 1-week follow-up visit, but the between-group difference in the change of right hemi-spheric PC-PHG functional connectivity disappeared at that timepoint. Therefore, we cannot exclude the possibility that the effect of theta-tACS on right hemispheric PC-PHG functional connectivity was a random finding. We also consider it necessary to exclude the possibility that the effect was purely an epiphenomenon from the impact of theta-tACS on the connectivity of other important large-scale networks implicated in schizophrenia due to the inherent limitations of the eLORETA seed-based approach for functional connectivity analysis with pre-defined seeds or ROIs”. In fact, the meaning of this paragraph is completely different from the conclusion stated now at abstract (lines 39-42) “Our findings suggest that in-phase theta-tACS can modulate theta-band large-scale functional connectivity pertaining to negative symptoms while right hemispheric PC-PHG functional connectivity may serve as a surrogate of treatment response.”
Author Response
Response to the reviewer
I would like thank the reviewer for the valuable and prompt comment. We revise our abstract in line 37-45 “We found that in-phase theta-tACS significantly reduced the LPS between the posterior cingulate (PC) and the parahippocampal gyrus (PHG) in the right hemisphere only at the end of stimulation relative to sham (p=0.0009, corrected). The reduction in right hemispheric PC-PHG LPS was significantly correlated with negative symptom improvement at the end of stimulation (r=0.503, p=0.039). Our findings suggest that in-phase theta-tACS can modulate theta-band large-scale functional connectivity pertaining to negative symptoms. Considering failure of right hemispheric PC-PHG functional connectivity to predict improvement in negative symptoms at one-week follow-up, future studies should investigate whether it can serve as a surrogate of treatment response to theta-tACS.”.